# Therapeutic Potential and Predictive Pharmaceutical Modeling of Stilbenes in *Cannabis sativa*

**DOI:** 10.3390/pharmaceutics15071941

**Published:** 2023-07-12

**Authors:** Conor O’Croinin, Andres Garcia Guerra, Michael R. Doschak, Raimar Löbenberg, Neal M. Davies

**Affiliations:** Faculty of Pharmacy and Pharmaceutical Sciences, Katz Centre for Pharmacy and Health Research, University of Alberta, Edmonton, AB T6G 2E1, Canada; ocroinin@ualberta.ca (C.O.); agarciag@ualberta.ca (A.G.G.); mdoschak@ualberta.ca (M.R.D.); raimar@ualberta.ca (R.L.)

**Keywords:** stilbene, bibenzyl, cannabis, anti-inflammatory, anti-cancer, in silico

## Abstract

*Cannabis sativa* is a plant used for recreational and therapeutic purposes; however, many of the secondary metabolites in the plant have not been thoroughly investigated. Stilbenes are a class of compounds with demonstrated anti-inflammatory and antioxidant properties and are present in cannabis. Many stilbenes present in cannabis have been investigated for their therapeutic effects. Fourteen stilbenes have been identified to be present in cannabis, all of which are structurally dihydrostilbenoids, with half possessing a prenylated moiety. The stilbenes summarized in this analysis show varying degrees of therapeutic benefits ranging from anti-inflammatory, antiviral, and anti-cancer to antioxidant effects. Many of the identified stilbenes have been researched to a limited extent for potential health benefits. In addition, predictive in silico modeling was performed on the fourteen identified cannabis-derived stilbenes. This modeling provides prospective activity, pharmacokinetic, metabolism, and permeability data, setting the groundwork for further investigation into these poorly characterized compounds.

## 1. Introduction

Current evidence suggests *Cannabis sativa* contains at least 550 different compounds [1], with new compounds continuously being isolated, identified, and characterized and potential new biomedical applications proposed. Cannabis contains a vast number of secondary metabolites; these are metabolites that are not crucial for the survival of the plant but can provide a selective advantage in the plant’s environment. Secondary metabolites such as cannabinoids and terpenes are now undergoing further investigation as potential lead compounds for further nutraceutical and pharmaceutical development. For example, it is well established that cannabinoids such as cannabidiol (CBD) and delta-9-tetrahydrocannabinol (delta-9-THC) provide therapeutic effects through direct or indirect modulation of the body’s naturally occurring endocannabinoid system [2]. The ongoing research of these compounds is justified by their numerous potential therapeutic and recreational uses, including analgesic, anti-inflammatory, and antioxidant effects [2]. Other secondary metabolites determined in *Cannabis sativa* have not been thoroughly investigated, including stilbenes and flavonoids. These compounds, some of which appear to be uniquely found in *Cannabis sativa*, also have the potential for similar or additional therapeutic benefits, but further characterization is required. The potential benefits of medicinal use of compounds such as stilbenes and flavonoids are evidenced by the therapeutic effects seen in similar compounds to the variants found in cannabis, with anti-cancer, antioxidant, and anti-inflammatory benefits [3,4]. Ongoing investigations have demonstrated that stilbenes, such as resveratrol, may have therapeutic applications and other secondary metabolites in cannabis, such as CBD and delta-9-THC, provide the rationale for further investigating stilbene compounds determined in cannabis for medicinal benefits. Investigating and characterizing potentially undetermined secondary metabolites such as stilbenes present in the plant is warranted. Further investigation and isolation of cannabis-derived compounds such as stilbenes will allow for their potential application as therapeutic agents.

Of the plethora of compounds already identified within cannabis, fourteen have been classified as stilbenes. Stilbene is a chemical constituent with characteristic dibenzal rings connected in either the E or Z configuration around a central double bond (Figure 1A,B) that leads to two geometric isomers. The Z stereoisomer often occurs more commonly in natural sources [5]. The hydroxylated derivative of stilbene is a stilbenoid, an example of which is the polyphenol structure of resveratrol (Figure 1C) that occurs in both the cis and trans conformations; however, this compound has not been identified in cannabis but in a related structure of which there have been extensive investigations, and it is available in many natural products and supplements. More specifically, the fourteen stilbenes determined so far in cannabis are dihydrostilbenoids (Figure 1D); this moiety is distinguished by the hydrogenation of a central double bond. Further modifications to the central dihydrostilbenoid structure differentiate the stilbenes found in cannabis from each other, with a common modification being prenylation. Prenylation is seen in half of the stilbenes determined in cannabis (Figure 2). The identified stilbenes discussed and analyzed herein are listed in Figure 2, and they are all bibenzyl compounds, as noted by the structural diagrams. Further derivations from the skeleton of the bibenzyl compounds are dihydrophenanthrenes and spirans [6].

## 2. Canniprene

Canniprene (Figure 3) is a prenylated dihydrostilbene compound found in cannabis and is among the first to have been isolated and identified. Canniprene was first isolated from a Thailand cannabis strain high in delta-9 THC, and the bibenzyl structure of canniprene was elucidated from this strain using nuclear magnetic resonance (NMR) [7]. Following the isolation of canniprene by Crombie et al. [7] in 1978, the same group reported the first synthesis of the compound two years later, making canniprene accessible for further research studies [8]. Canniprene is not unique to the strain of Thailand cannabis, with further research identifying it again in a Panamanian strain of high delta-9 THC content [9]. More recent research suggests that canniprene concentration in a plant shows no distinct relationship with the specific cannabinoid profile present or the age of the plant [10]. The relationship between canniprene concentration and the concentration of cannflavin A and B, similar polyphenol compounds found in cannabis, was also investigated by Allegrone et al. [10], with an inverse relationship being identified. This suggests a competition between the biosynthesis pathways of these polyphenol compounds [10].

### 2.1. Anti-Inflammatory Effects 

A major therapeutic effect that canniprene demonstrates is the anti-inflammatory nature of the compound. The investigation of canniprene as an anti-inflammatory stems from prior research into the related polyphenol structures of cannflavin A and B, which demonstrated inhibition of the 5-lipoxygenase (5-LO) and microsomal prostaglandin E_2_ synthase (mPGES) pathways [11]. This foundational research provided the outcomes upon which Allegrone et al. [10] measured the anti-inflammatory activity of canniprene. The 5-LO pathway is a metabolic pathway for arachidonic acid breakdown, which results in the production of inflammatory eicosanoids [12]. Similar to cannflavin A and B, the production of these eicosanoids is inhibited by canniprene [10]. Canniprene is a potent inhibitor of 5-LO, which is an enzyme necessary for the inflammatory response of this pathway. 5-LO inhibition by canniprene was seen in a concentration-dependent manner, outperforming the 5-LO inhibitor medication zileuton [10]. Inhibition of this pathway has potential health benefits in regard to the treatment of asthma, allergic rhinitis, COPD, and idiopathic pulmonary fibrosis, in which the pathway is implicated [12].

The mPEGS pathway is responsible for a large quantity of prostaglandin E_2_ production during inflammation, with mPEGS-1 catalyzing the final step in the biosynthesis of prostaglandin E_2_ from arachidonic acid [13]. Where traditional non-steroidal anti-inflammatory drugs are commonly associated with side effects with chronic use, mPEGS-1 inhibition lacks gastrointestinal, renal, and cardiovascular adverse effects and thus is potentially beneficial to those with chronic inflammation [13]. Canniprene shows concentration-dependent inhibition of mPEGS-1; however, the effects are not as potent as what was seen in the inhibition of the 5-LO pathway, with canniprene also being less potent than cannflavin A in terms of mPEGS-1 inhibition [10].

The relationship between canniprene and cyclooxygenase-1 (COX1) and cyclooxygenase-2 (COX2) was investigated using in silico modeling by Romero Salamanca and Castañeda Castellanos [14]. The inhibition constant determined for canniprene on COX1 is 14.53 µM, and for COX2 is 0.21 µM [14]. The inhibition of COX2 is anti-inflammatory due to the decreased production of prostaglandins, and the more potent inhibition of COX2 rather than COX1 can be beneficial as COX1 inhibition can lead to gastric ulcerations [15]. These results provide strong evidence for further investigation into the use of canniprene as an anti-inflammatory agent. 

### 2.2. Anti-Cancer Effects

The capability of canniprene to act as an anti-cancer agent was investigated by Guo et al. [16], who tested canniprene on the cancer cell lines MCF-7 (human breast cancer), A549 (human lung adenocarcinoma), HepG2 (human liver carcinoma), and HT-29 (human colorectal adenocarcinoma). The antiproliferative actions of canniprene against these cancer types were determined, with very strong effects seen against all four cell lines [16]. The inhibition of A549 proliferation was slightly lower than the other three cell lines, but all remained above an 80% cell inhibitory rate. Guo et al. [16] also tested the cytotoxic effects of canniprene on the cancer cells and determined the same results, with cytotoxicity approaching or greater than 90% cell death for all four cancer types. These data suggest that canniprene has potential therapeutic benefits as an antitumor agent.

### 2.3. Antiviral Effects

The COVID-19 pandemic has brought into focus the severe consequences of a lack of antiviral therapies for diseases such as COVID-19. The SARS-CoV-2 virus has necessitated the investigation of a diverse range of possible treatments and preventative measures to end the pandemic, including phytoligands in cannabis. Canniprene is one of 29 phytoligands investigated by Khattab and Teleb [17] using in silico modeling for a possible interaction with either the SARS-CoV-2 binding site or the viral replication pathway. SARS-CoV-2 uses spike glycoproteins to bind to angiotensin-converting enzyme 2 (ACE2); disrupting this SARS-CoV-2-ACE2 complex is potentially beneficial in preventing viral infection [17]. Due to the endogenous processes the ACE2 receptor is involved in, ligands that do not disrupt the ACE2 receptor would be the most beneficial. However, no interactions between canniprene and the SARS-CoV-2-ACE2 complex were observed [17]. Canniprene did show very strong antiviral activity through the disruption of virus replication. The main protease (Mpro) is a crucial component in the development of viral nonstructural proteins and is the target of the antiviral activity of canniprene. In silico modeling demonstrates that canniprene inhibits Mpro and has the highest binding affinity of the 29 phytoligands from cannabis that were examined, including flavonoids and cannabinoids [17]. This inhibition of SARS-CoV-2 replication demonstrated by in silico modeling provides the foundation for further examination of canniprene as a potential treatment for the virus.

## 3. Cannabistilbene

Another stilbene found in cannabis is cannabistilbene. This compound is present in three forms: cannabistilbene I, cannabistilbene IIa, and cannabistilbene IIb. Of the three structures, only cannabistilbene I is prenylated. There is limited research into cannabistilbene since its initial isolation almost forty years ago by ElSohly et al. [18], who extracted cannabistilbene from a Panamanian strain of cannabis. This initial discovery of the molecule was characterized using spectral data to elucidate the structures, then synthesis of the proposed structures to confirm the spectral data of the extracted compounds [18]. ElSohly et al. [18] were able to determine the structure of cannabistilbene II as one of two isoforms (Figure 4C,D). The initial NMR spectra suggested the possibility of a symmetrical distribution of the substituents seen in Figure 4B, but a comparison of the spectra with spectra from similar compounds, such as canniprene, allowed for the determination that the functional groups are ortho-substituted leading to the two possible structures of cannabistilbene II [18]. Further research must be conducted to determine which is actually present in cannabis. 

The biosynthesis of cannabistilbene I and II follow different pathways, as made apparent by the prenylation of I but not IIa or IIb. Evidence suggests cannabistilbene I is derived from another stilbene in cannabis, dihydroresveratrol [19]. From dihydroresveratrol, methylation is required, followed by prenylation. Dihydroresveratrol is methylated by the cannabis methyltransferase CsOMT1 and further prenylated by the soluble prenyltransferase CloQ [19]. The synthesis of cannabistilbene IIa and IIb, on the other hand, originates from dihydro-p-coumaroyl-CoA, the same compound from which dihydroresveratrol is produced [20,21]. Limited research has been published regarding the therapeutic properties of cannabistilbene I or II; however, the benefits of the related stilbenes highlight the importance of this research in the future. 

### Anti-Inflammatory Effects

The same in silico modeling that was performed in canniprene was also investigated with the three forms of cannabistilbene to determine the inhibitory effects of the compounds on COX1 and COX2 [14]. For the inhibition of COX1, cannabistilbene IIb was the most potent, with an inhibition constant of 1.37 µM, followed by cannabistilbene I at 2.27 µM and then cannabistilbene IIa at 14.53 µM [14]. Cannabistilbene I and cannabistilbene IIa were determined to be very strong inhibitors of COX2 as both had an inhibition constant of 0.21 µM whereas cannabistilbene IIb was not as potent with a value of 2.69 µM [14]. The comparatively stronger inhibition of COX2 over COX1 seen in cannabistilbene I and IIa is beneficial as COX2 inhibition is anti-inflammatory, while COX1 inhibition has shown negative gastric side effects [15]. 

## 4. 3,4′-Dihydroxy-5,3′-dimethoxy-5′-isoprenylbibenzyl

The next prenylated stilbene found in cannabis, 3,4′-dihydroxy-5,3′-dimethoxy-5′-isoprenylbibenzyl (Figure 5), also has limited research related to it. The structure of the compound was first elucidated by Kettenes-van den Bosch and Salemink [22], who confirmed the structure using MS, IR, and NMR spectral data. The authors found no evidence of 3,4′-dihydroxy-5,3′-dimethoxy-5′-isoprenylbibenzyl in the smoke condensate of the strain of Mexican cannabis they investigated, suggesting that dihydrostilbene could be transformed into a derivative during thermolysis, or the concentration was too low to be detected. This Mexican strain of cannabis showed very low concentrations (0.01%) of 3,4′-dihydroxy-5,3′-dimethoxy-5′-isoprenylbibenzyl; however, further research must be conducted into the concentrations of the compound in other strains of cannabis [22]. The therapeutic potential of this compound has largely not been investigated, but the benefits of related compounds provide justification for further research in this area. Care should be taken when researching this compound, as some authors have mistakenly drawn an isomer of this compound while referring to it by the same name.

### Anti-Inflammatory Effects

Romero Salamanca and Castañeda Castellanos [14] also investigated the anti-inflammatory effects of 3,4′-dihydroxy-5,3′-dimethoxy-5′-isoprenylbibenzyl through in silico modeling of COX1 and COX2 inhibition. The inhibition constant for this compound was 6.25 µM for COX1 and 0.35 µM for COX2. This modeling predicts the strong inhibitory effect of this stilbene on both COX1 and COX2. Despite the inhibition of COX1, the inhibition of COX2 is beneficial for preventing inflammatory prostaglandin production and is mechanistically similar to non-steroidal anti-inflammatory drugs (NSAIDs) [15]. This in silico modeling can guide future studies using in vitro or in vivo methodology. 

## 5. HM1, HM2, and HM3

α,α′-dihydro-3′,4,5′-trihydroxy-4′-methoxy-3-isopentenylstilbene (HM1), α,α′-dihy- dro-3,4′,5-trihydroxy-4-methoxy-2,6-diisopentenylstilbene (HM2),and α,α′-dihydro-3′, 4,5′-trihydroxy-4′-methoxy-2′,3-diisopentenylstilbene (HM3) (Figure 6) were first identified by Guo et al. [16]. The authors determined the structures of these three compounds using LC-MS, H-NMR, C-NMR, and IR spectra. The three novel dihydrostilbenes were isolated and extracted from the leaves of a strain of cannabis from the Yunnan province of China.

### 5.1. Anti-Cancer Effects

The potential benefits of HM1, HM2, and HM3 for treating different cancers were tested in vitro by Guo et al. [16]. The cell lines MCF-7 (human breast cancer), A549 (human lung adenocarcinoma), HepG2 (human liver carcinoma), and HT-29 (human colorectal adenocarcinoma) were used to test the efficacy of these compounds at inhibiting cancer cell growth. HM1 showed selective inhibition with very strong inhibition seen for the MCF-7 and A549 cell lines and limited inhibition of HepG2 and HT-29 [16]. HM2 and HM3 did not show the same degree of growth inhibition towards MCF-7 and A549. HM2 showed the highest inhibition towards MCF-7 and only mild inhibition of growth in the other cell lines. HM3 showed the same selective inhibition of cell proliferation to MCF-7 and A549 as HM1 but was not as effective [16]. It was demonstrated that there is not only a prevention of cancer cell proliferation but also a cytotoxic effect towards the cancer cell lines. These results suggest that HM1, HM2, and HM3 have potential benefits as antitumor therapies for cancer, especially breast and lung cancer [16].

### 5.2. Cardioprotective Effects

In addition to testing the efficacy of HM1, HM2, and HM3 as anti-cancer agents, Guo et al. [16] tested the impact these novel compounds have on cholesterol transporters. ABCG1 and SR-B1 are cholesterol transporters crucial for the efflux of cholesterol from cells and the transformation into high-density lipoprotein (HDL) cholesterol. When tested, HM1, HM2, and HM3 all increased the protein expression of these transporters, demonstrating a cardioprotective mechanism of action [16]. In addition to these two transporters, ABCA1 was also investigated as a protein responsible for transporting cholesterol to apolipoprotein A-1 which is a major component of HDL [23]. The three novel dihydrostilbenes all increased the expression of ABCA1 [16]. An increase in HDL cholesterol, as these results indicate, is beneficial as there is an inverse relationship between levels of HDL cholesterol and the risk of atherosclerosis [24]. Reverse cholesterol transport (RCT) is the removal of cholesterol from peripheral cells to the liver, where it is catabolized and excreted [23]. The above evidence from Guo et al. [16] demonstrates an increased ability to initiate RCT. The authors also tested HM1, HM2, and HM3 on HepG2 cells to determine the impact in the liver further along the RCT pathway. They found that the three dihydrostilbenes increase the protein expression of SR-B1 in hepatocytes, and HM1 and HM3 promote cholesterol uptake through an indirect method of transport [16]. A key enzyme for converting cholesterol to bile acid in the liver is CYP7A1 which showed increased expression after exposure to HM1 and, to a lesser extent, HM2 and HM3 [16]. These results further demonstrate the cardioprotective potential of these compounds to limit atherosclerotic heart disease.

## 6. α,α′-Dihydro-3,4′,5-trihydroxy-4,5′-diisopentenylstilbene

The prenylated dihydrostilbene α,α′-dihydro-3,4′,5-trihydroxy-4,5′-diisopenten- ylstilbene (Figure 7) was first identified in cannabis by Guo et al. [16]. The compound had previously been identified, but in Glycyrrhiza glabra, otherwise known as liquorice [25]. The structure of α,α′-dihydro-3,4′,5-trihydroxy-4,5′-diisopentenylstilbene was confirmed in liquorice using MS, NMR, and IR spectra.

### 6.1. Anti-Cancer Effects

Fifteen years after α,α′-dihydro-3,4′,5-trihydroxy-4,5′-diisopentenylstilbene was found in liquorice, it was identified in cannabis by Guo et al. [16] who were largely interested in the ability of the stilbenes to influence cancer cell proliferation. They tested this compound on the human cancer cell lines MCF-7, A549, HepG2, and HT-29. They determined that the stilbenes HM1, HM2, and HM3 showed selective inhibition of only MCF-7 and A549, but α,α′-dihydro-3,4′,5-trihydroxy-4,5′-diisopentenylstilbene did not show any selectivity. This compound inhibited the proliferation of all four cancer types at inhibition rates of above 90% [16]. After it was determined that the dihydrostilbene inhibits cancer cell growth, the cytotoxicity towards these cell lines was also tested. Guo et al. [16] demonstrated that α,α′-dihydro-3,4′,5-trihydroxy-4,5′-diisopentenylstilbene was toxic to all four cancer types. These data suggest that there is a potential therapeutic benefit for the use of α,α′-dihydro-3,4′,5-trihydroxy-4,5′-diisopentenylstilbene as an antitumor agent for breast cancer, lung adenocarcinoma, liver carcinoma, and colorectal adenocarcinoma therapy. 

### 6.2. Antioxidant Effects

Before α,α′-dihydro-3,4′,5-trihydroxy-4,5′-diisopentenylstilbene was identified in cannabis, the compound was tested for its efficacy as an antioxidant. The absolute inhibition of oxidation rate constants for the sample extracted from liquorice was determined [25]. The rate constant for α,α′-dihydro-3,4′,5-trihydroxy-4,5′-diisopentenylstilbene of 70 k (M^−1^s^−1^) allowed for the compound to be classified as a good antioxidant, close to the 100–1000 k (M^−1^s^−1^) range seen in very effective antioxidants [25]. 

## 7. Dihydroresveratrol

The most well-studied stilbene in cannabis is dihydroresveratrol (Figure 8), a dihydrostilbene that is not prenylated. The increased investigation of dihydroresveratrol is largely due to the fact that it is a phase I hydrogenated metabolite of resveratrol, a compound also metabolized in humans by gut bacteria [26]. Resveratrol is a stilbene determined in many natural sources, including wines, is available in many nutraceutical products, and it is well understood to have anti-inflammatory, anti-cancer, and antioxidant activity, thus providing the rationale for the investigation of the same properties in dihydroresveratrol [5]. There has been no evidence that resveratrol is found in cannabis, whereas dihydroresveratrol was discovered in cannabis by El-Feraly [27], who identified it in a sample of cannabis from India. This suggests that dihydroresveratrol biosynthesis occurs from a different precursor than resveratrol in cannabis. The biosynthesis pathway was recently elucidated by Boddington et al. [20], who provided evidence that the pathway originates from the hydroxycinnamic acid known as p-coumaric acid. The first step of this pathway is the esterification of p-coumaric acid with coenzyme A (CoA), followed by reduction to create dihydro-p-coumaroyl-CoA, and finally, the formation of the bibenzyl ring and the complete dihydroresveratrol structure using the enzyme CsBBS2 [20]. Methods for investigating dihydroresveratrol have also been established, including a validated method for detecting the compound using high-performance liquid chromatography (HPLC) after extraction from rat plasma, laying the foundation for future studies using rat animal models [28]. The isolation and detection of a dihydroresveratrol metabolite from human urine have been accomplished, providing further studies with a potential method for quantifying the compound in human trials [29]. Dihydroresveratrol may also alter the metabolism of other compounds through interactions with cytochrome P450 enzymes in the liver, as it shows slight inhibition of CYP2C19, CYP2D6, and CYP3A4 enzymes [30]. The metabolic pathways dihydroresveratrol is subjected to in a rat were recently determined by Ji et al. [31], who used LCMS data from rat urine, hepatocytes, and bile to identify 16 metabolites. During phase I metabolism, dihydroresveratrol undergoes hydroxylation and dehydrogenation, and in phase II metabolism, glucuronidation, GSH conjugation, and methylation occur [31]. 

### 7.1. Anti-Inflammatory Effects

Due to the well-accepted anti-inflammatory actions of resveratrol, the potential for dihydroresveratrol to also be anti-inflammatory has been investigated [5]. Interleukins are a class of cytokines responsible for a proinflammatory response, and by measuring changes in gene expression of these cytokines in response to dihydroresveratrol, an anti-inflammatory effect can be investigated. Dihydroresveratrol was shown to decrease the mRNA expression of IL-6 [32]. IL-6 is a proinflammatory cytokine that is involved in the inflammatory immune response but can also play a factor in chronic inflammation and autoimmune diseases [33]. Zhang et al. [32] also provided in vitro evidence that dihydroresveratrol downregulates the expression of IL-1β and IL-18, two members of the Interleukin 1 family. Inflammatory cytokines in the IL-1 family are linked to the pathological mechanisms of diseases such as metabolic syndromes, autoinflammatory diseases, and acute and chronic inflammation [34]. The downregulation of these cytokines by dihydroresveratrol during in vitro experiments by Zhang et al. [32] was further expanded in a mouse model of colitis where, of the interleukins tested, only IL-1β had decreased serum concentrations. The anti-inflammatory actions Zhang et al. [32] presented for dihydroresveratrol with interleukins were accompanied by a proinflammatory response in the form of TNF-α. The authors found that during in vitro experiments, the expression of TNF-α, a proinflammatory cytokine, was increased [32]. Taken together, these results suggest that although the effect of dihydroresveratrol on IL-6 and IL-1 family cytokines is beneficial towards a net anti-inflammatory response, the concurrent increase in TNF-α expression has the potential to offset the anti-inflammatory actions seen. Additional research using the colitis mouse model demonstrated that a higher ratio of dihydroresveratrol to resveratrol led to a greater anti-inflammatory response [35]. The authors determined that dihydroresveratrol was eliciting this effect through the upregulation of the aryl hydrocarbon receptor, which promotes intestinal integrity and prevents inflammation. The same study found that dihydroresveratrol downregulates intestinal 5HT7 receptors, which are upregulated in colitis, providing a potential biomarker for the efficacy of colitis amelioration [35]. Another commonly studied inflammatory mediator is nitric oxide, which is not impacted by dihydroresveratrol in an in vitro model unless it is supplemented with dihydrostilbene lunularin [36]. In addition to the aforementioned inflammatory pathways, there is evidence from in silico modeling that dihydroresveratrol inhibits COX1 (inhibition constant of 3.77 µM) and COX2 (inhibition constant of 1.37 µM) [14]. COX2 inhibition is the mechanism through which NSAIDs elicit their anti-inflammatory effects, so these results support the possibility of a similar method of action for dihydroresveratrol. Further research to elucidate the impact of dihydroresveratrol on inflammation in humans seems clinically prudent. 

### 7.2. Anti-Cancer Effects

The potential for polyphenol compounds to act as antiproliferative agents towards breast cancer cell lines was investigated by Ávila-Gálvez et al. [37], who aimed to elucidate the effects of phase II metabolism on previously demonstrated in vitro findings. The investigators hypothesized that phase II metabolism following absorption of dietary polyphenols such as dihydroresveratrol would limit the anti-cancer effects that the compounds demonstrate. They tested dihydroresveratrol on the breast cancer cell lines MCF-7 (estrogen receptor positive) and MDA-MB-231 (estrogen receptor negative) to determine whether there is a decrease in cell proliferation following exposure to polyphenols. It was determined that dihydroresveratrol decreases the rate of proliferation in MDA-MB-231 cells at high concentrations but has no impact at low concentrations on MDA-MB-231 and no effect on MCF-7 proliferation [37]. The results seen in MCF-7 cells are contrary to a prior study which showed potent increases in the proliferation of these cancer cells in response to dihydroresveratrol at low concentrations and cytotoxicity towards the cell line at high concentrations [38]. Similar to the results of Ávila-Gálvez et al. [37], data from Gakh et al. [38] demonstrated cytotoxicity towards MDA-MB-231 cancer at high concentrations of dihydroresveratrol. The same data were obtained by Ávila-Gálvez et al. [37] for the compound dihydroresveratrol-3-O-glucuronide, a phase II metabolite of dihydroresveratrol. This glucuronide conjugation eliminated the anti-cancer effect seen in the MDA-MB-231 cells at high concentrations, providing support for the hypothesized effect that phase II metabolism has on dihydroresveratrol’s anti-cancer activity [37]. A follow-up study investigated the impact dihydroresveratrol and dihydroresveratrol-3-O-glucuronide have on long-term anti-cancer activity rather than the short-term effects previously investigated [39]. A clonogenic assay demonstrates the ability of a compound to prevent a cell from replicating to form a colony over a 12-day period. This assay showed that at a concentration of 10 µmol/L, dihydroresveratrol-3-O-glucuronide but not dihydroresveratrol prevents the formation of the colony, with a 39.1% decrease in colony formation in the conjugated compound [39]. This suggests dihydroresveratrol-3-O-glucuronide limits breast cancer cell proliferation, unlike the conclusions of Ávila-Gálvez et al. [37], with the discrepancy being understood through the longer exposure time used by Giménez-Bastida et al. [39]. Interestingly, the authors also found that a 1 µmol/L concentration of dihydroresveratrol prevented colony formation despite the lack of activity at the higher concentration. To determine the mechanism of this long-term clonogenic inhibition by dihydroresveratrol and the conjugated version of the compound, the ability of the compound to induce cellular senescence was examined [39]. The authors used senescence-associated β-Galactosidase as a marker for the induction of senescence by the compounds and demonstrated a clear increase in senescence after 3 days and 5 days of dihydroresveratrol-3-O-glucuronide exposure and after 3 days but not 5 days of dihydroresveratrol exposure [39]. This suggests that inducing cellular senescence could contribute to the potential long-term benefits of dihydroresveratrol in preventing breast cancer. The mechanism for this induction of senescence was determined to be the regulation of the expression of proteins responsible for the progression of the cell cycle, such as p53 and p21 [39]. The effect of dihydroresveratrol on the cellular senescence of human fibroblasts was also investigated in the context of potential anti-aging benefits, but the results showed no significant effect [40]. The antiproliferative and anti-clonogenic effects of dihydroresveratrol were tested for colonic and renal cancer cell lines as well, but no significant effect was seen for either cell line unless dihydroresveratrol was combined with another dihydrostilbene, lunularin [36]. The effect dihydroresveratrol has on prostate cancer has also been investigated, and a biphasic response similar to that found by Gakh et al. [38] in breast cancer was identified [41]. It was demonstrated that at picomolar concentrations, dihydroresveratrol promotes the proliferation of the cancer cells, but at higher concentrations, the expected anti-cancer effects are seen as proliferation being inhibited [41]. 

### 7.3. Antioxidant Effects

Acute pancreatitis is an inflammatory disease that causes premature activation of proteolytic zymogens, causing autodigestion of the pancreatic exocrine cells [42]. Although the exact cause of acute pancreatitis is still largely unknown, it is well-accepted that oxidative stress plays a role in the initiation of the inflammatory response, with an overproduction of reactive oxygen species causing damage [42]. This justifies the use of antioxidants as a potential treatment for acute pancreatitis. Recent evidence has provided a mechanism that shows dihydroresveratrol is capable of quenching singlet oxygen, a common reactive oxygen species [43]. Tsang et al. [42] investigated the effect dihydroresveratrol has as an antioxidant in rat models of acute pancreatitis. The authors demonstrated that dihydroresveratrol was effective at reducing the severity of acute pancreatitis in the rats, as seen by lowered levels of edema, plasma α-amylase, and pancreatic leukocyte infiltration [42]. To determine the antioxidant mechanism of dihydroresveratrol, Tsang et al. [42] evaluated the NADPH oxidase system, a common source of the reactive oxygen species. They found that rats treated with dihydroresveratrol had a significant decrease in NADPH oxidase activity and a decrease in nuclear factor kappa B (NF-κB) expression. NF-κB regulates the transcription of proinflammatory cytokines and is upregulated by reactive oxygen species from the NADPH oxidase system [42]. These findings were supported by Lin et al. [44], who also demonstrated this decrease in NF-κB activation and showed that dihydroresveratrol lessens the injury to the lungs of rats in this acute pancreatitis model. The NADPH oxidase and NF-κB relationship is a link between oxidative stress and the inflammatory response seen in acute pancreatitis and outlines the pathway that is impacted by dihydroresveratrol administration. Tsang et al. [42] concluded that dihydroresveratrol should be further investigated as a therapeutic agent for acute pancreatitis.

### 7.4. Anti-Diabetic Effects

Fatty acid binding protein 4 (FABP4) is a lipid chaperone that contributes to insulin resistance, with high circulating levels of FABP4 being associated with type II diabetes [45]. Decreasing FABP4 levels is also associated with augmented cardiomyocyte pathology and improved ventricular function in diabetic mouse models, with the increased FABP4 levels seen in type II diabetes potentially the cause of diabetic-induced cardiomyopathy [45]. Azorín-Ortuño et al. [46] investigated the impact dihydroresveratrol has on FABP4 levels in peripheral blood mononuclear cells. The authors were able to demonstrate a small but statistically significant reduction in the concentrations of FABP4 in response to dihydroresveratrol at low micromolar concentrations [46]. These findings suggest a potential use of dihydroresveratrol in preventing diabetic-induced cardiomyopathy. The integrity of the intestinal tight junctions is crucial for gut homeostasis and determining the permeability of the intestinal epithelial cells to different solutes [47]. Intestinal barrier dysfunction is linked to the pathological autoimmune response in type I diabetes, indicating why it is beneficial to maintain the integrity of the intestinal wall [48]. The relationship between dihydroresveratrol and intestinal dysfunction has been investigated, but despite restoration of tight junction proteins by resveratrol, no such change was observed in response to dihydroresveratrol [47]. Caloric restriction is a well-known principle to protect against the development or progression of type II diabetes [49]. The ability to promote caloric restriction through the intake of compounds such as dihydroresveratrol would therefore have anti-diabetic effects. Pallauf et al. [50] injected mice intraperitoneally with dihydroresveratrol and compared these mice to calorie-restricted mice. They demonstrated that the weight loss, changes in blood proteins, and changes in genetic markers seen in the calorie-restricted mice were not seen following the injection of dihydroresveratrol, concluding that it does not mimic caloric restriction [50]. These results were supported by Günther et al. [51], who found no hormone or metabolome changes that would be associated with caloric restriction in mice given dihydroresveratrol. The authors also performed an α-glucosidase assay and found dihydroresveratrol was able to inhibit α-glucosidase, but not with the same potency as resveratrol [51]. This α-glucosidase inhibition is potentially beneficial for diabetics because inhibiting this enzyme lowers blood glucose levels by delaying the absorption of carbohydrates [51]. It is estimated that 75% of patients with type II diabetes suffer from hepatic steatosis, or fatty liver disease, which can lead to increased mortality for these patients [52]. Preventing the accumulation of fat in the liver would be beneficial to healthy and diabetic populations, and the efficacy of dihydroresveratrol in fighting this disease has been investigated. Dihydroresveratrol is able to decrease fat accumulation in mouse hepatocyte cells, providing potential preclinical support for its use in treating steatosis [53]. To further determine the overall impact of dihydroresveratrol on diabetes, studies with human participants afflicted with diabetes should be performed as a potential add-on therapy for type II diabetes [54].

## 8. 3,4′-Dihydroxy-5-methoxy Bibenzyl

The non-prenylated dihydrostilbene 3,4′-dihydroxy-5-methoxy bibenzyl (Figure 9) was first identified by Kettenes-van den Bosch and Salemink [22] and isolated from a strain of Mexican cannabis. 3,4′-dihydroxy-5-methoxy bibenzyl was detected in very low amounts in the cannabis extract and was not detectable in the smoke condensate, potentially due to structural changes during thermolysis [22]. The structure of this stilbene was determined using MS, IR, and NMR spectra and confirmed by molecular synthesis.

### 8.1. Anti-Inflammatory Effects

Lipopolysaccharide-induced macrophages can simulate an inflammatory environment in vitro. Su et al. [55] used this model to determine the anti-inflammatory effects of 3,4′-dihydroxy-5-methoxy bibenzyl. There was significant inhibition of nitric oxide, TNF-α, and IL-1β after the application of 3,4′-dihydroxy-5-methoxy bibenzyl to the macrophages [55]. The inhibition of these inflammatory mediators is beneficial for the resolution of chronic inflammatory diseases and metabolic syndromes in which these mediators are implicated [34]. Similar to other stilbenes, the anti-inflammatory properties of 3,4′-dihydroxy-5-methoxy bibenzyl to inhibit COX1 and COX2 were determined through in silico modeling [14]. These data predict that 3,4′-dihydroxy-5-methoxy bibenzyl inhibits COX1 with an inhibition constant of 10.37 µM and COX2 with an inhibition constant of 1.62 µM [14]. This potential COX inhibition would limit the production of prostaglandins, key mediators in the inflammatory response [15]. Further research to determine whether these anti-inflammatory effects translate to other models of inflammation and potential clinical effects is required.

### 8.2. Estrogenic Effects 

The effects that 3,4′-dihydroxy-5-methoxy bibenzyl has on estrogen production have been investigated in animal models. The cannabis plant has established estrogenic effects, and Wirth et al. [56] aimed to delineate the compounds within cannabis that contribute to these effects. The Sprague Dawley rat model and the uterine weight of female prepubescent rats were utilized as markers of estrogenic activity. A significant increase in the uterine weight in response to the addition of 3,4′-dihydroxy-5-methoxy bibenzyl at doses of 50 mg/kg and 250 mg/kg was evident [56]. These findings suggest that 3,4′-dihydroxy-5-methoxy bibenzyl is estrogenic, but the degree of activity did not account for all the estrogenic effects seen in cannabis, suggesting the involvement of other compounds in cannabis [56].

## 9. Gigantol

The compound 3,3′-dihydroxy-5,4′-dimethoxy bibenzyl (gigantol) is a non-prenylated stilbene having been determined in cannabis (Figure 10A). It was first identified in cannabis by Kettenes-van den Bosch and Salemink [22], who isolated it from a strain of Mexican cannabis. The structure of gigantol was confirmed using NMR, IR, and MS spectra, with the compound present in low concentrations but undetectable in the smoke condensate of the cannabis strain [22]. The therapeutic potential of gigantol from cannabis is well studied; however, discrepancies arise in the literature regarding the actual structure of the compound and the use of the moniker “gigantol”. A positional isomer (Figure 10B) of the compound gigantol seen in cannabis (Figure 10A) is commonly present in orchids and is very well studied [57]. This positional isomer has also been referred to as “gigantol” in many publications, but it is chemically 3,4′-dihydroxy-5,5′-dimethoxy bibenzyl which has demonstrated anti-cancer [58], antinociceptive [59], anti-inflammatory [59,60], antioxidant [61], and spasmolytic effects [62] among others; however, there is no evidence that this positional isomer is actually present in cannabis. Further confusion arises in the literature with the use of “gigantol” to also describe 4,4′-dihydroxy-3,3′,5-trimethoxy bibenzyl (Figure 10C), which is structurally related but not an isomer of the cannabis-derived gigantol. The conflicting nomenclature for these different stilbenes requires investigators to perform both MS and NMR identification and analysis of the “gigantol” obtained and utilized for research purposes ab initio in any investigations. The health benefits seen in the structurally similar stilbenes provide the rationale for investigating the benefits of cannabis “gigantol” further. 

### 9.1. Anti-Inflammatory Effects

The gigantol found in cannabis has been demonstrated to have anti-inflammatory effects. Using in silico modeling, the interaction between gigantol and COX1 and COX2 was investigated [14]. Gigantol has strong inhibitory effects against COX1 and COX2 based on modeling data, with an inhibition constant of 2.27 µM for COX1 and 1.37 µM for COX2 [14]. Although the inhibition of COX1 is associated with gastric ulceration, the subsequent inhibition of COX2 is well-established to be anti-inflammatory through the inhibition of prostaglandin formation, with NSAIDs working through this pathway [15]. These modeling data provide the foundation for further studies using in vitro or in vivo models. 

### 9.2. Anti-Cancer Effects

The gigantol isomer found in cannabis has also demonstrated anti-cancer effects. Gigantol has been shown to decrease the proliferation of breast cancer cells, including the cell lines MDA-MB-468 and MCF-7, in a concentration-dependent manner [63]. The ability of gigantol to enhance the cytotoxic effects of the chemotherapeutic agent Cisplatin on breast cancer cells was investigated [63]. The co-treatment of Cisplatin with gigantol significantly increased the toxicity towards the cell lines, with a change in IC_50_ of 12.29 μM to 2.92 μM for MDA-MB-468 and 22.66 μM to 8.36 μM for MCF-7 with the addition of 60 μM gigantol to the Cisplatin [63]. Gigantol also increased the apoptosis that both breast cancer cell lines experience with the addition of Cisplatin [63]. The effectiveness of gigantol as an anti-cancer agent against lung cancer has also been investigated. The compound has demonstrated the ability to suppress lung cancer stem cells which are important for tumor formation and regrowth, proving effective at limiting cancer stem cell proliferation and survival [64,65]. Gigantol significantly decreased the size and density of lung cancer tumors in a xenograft model and decreased tumor integrity [65]. The mechanism of action of gigantol to stop lung cancer cell growth is related to the downregulation of MYC, a proto-oncogene and transcription factor that promotes cancer cell proliferation [66]. Another important target of gigantol in lung cancer is the mesenchymal–epithelial transition factor, which is a tyrosine kinase that aids in the metastasis of cancer and is inhibited by gigantol [67]. Gigantol has also demonstrated the ability to prevent the proliferation of hepatocellular carcinoma cells and prevent migration and metastasis of these cells [68]. Gigantol can prevent the proliferation of cervical cancer as well, with evidence suggesting it does this through the initiation of oxidative stress in these cancer cells, increasing apoptosis and decreasing proliferation through the formation of reactive oxygen species in the mitochondria [69]. These results, taken together, provide strong evidence for the use of gigantol as an anti-cancer agent against breast, lung, liver, and cervical cancers. 

### 9.3. Antioxidant Effects 

The synthetic radical DPPH is used as a model for determining the ability of a compound to scavenge for free radicals due to the visible color change that can be measured with a spectrophotometer. Hydroxyl radicals, on the other hand, are the most toxic naturally occurring radicals, and the ability of a compound to scavenge them can also be measured. Ahammed et al. [70] used both these models to investigate the antioxidant capability of the gigantol isomer found in cannabis. They found gigantol to be a potent inhibitor of DPPH and hydroxyl radicals, showing greater antioxidant activity than all other tested compounds [70]. These assays set a strong foundation for gigantol as an antioxidant, but further research is needed.

### 9.4. Spasmolytic Effects

Hernández-Romero et al. [62] were primarily interested in the effect orchid-derived gigantol has on relieving smooth muscle spasms in the ileum, but they also tested the gigantol isomer that is present in cannabis for the same spasmolytic potential. The animal model they tested demonstrated that the gigantol isomer found in cannabis is effective at relieving smooth muscle spasms in the ileum of the rat. The mechanism of action for these observed effects was elucidated through the investigation of the compound’s effect on calmodulin, a calcium-binding protein crucial for smooth muscle contraction. It was confirmed that gigantol inhibits calmodulin, even to the extent that it showed greater potency than the positive control chlorpromazine, a noted calmodulin inhibitor [62]. The spasmolytic effect of gigantol could be beneficial in treating irritable bowel syndrome, as antispasmodics are a common treatment for this condition [71].

### 9.5. Androgenic Effects

The production of progesterone occurs in the Leydig cells of the male testes, where the sex hormone is derived from cholesterol, and the production can be impacted by gigantol [72]. The structure of gigantol tested by Basque et al. [72] matches the isomer that is present in cannabis. The results show that gigantol increases the expression of genes responsible for cholesterol and steroid production when the compound is applied to Leydig cells [72]. This overexpression of cholesterol-producing genes was shown to have a significant effect on the production of progesterone. These findings suggest gigantol could delay age-related hypogonadism, but further research in animal models should be undertaken [72].

### 9.6. Antiparasitic Effects

Trypanosomatid parasites are the cause of the disease leishmaniasis, which can be transmitted to humans through the bites of sandflies found in subtropical regions of the world [73]. Leishmaniasis causes around 70,000 deaths per year, and symptoms can include rashes, anemia, fevers, and hepatomegaly [73]. In silico molecular docking was used to determine the effectiveness of gigantol at binding to sterol 24-C methyltransferase, an enzyme in the parasite crucial for its viability while also unique to the parasite and not found in humans, making it an ideal target for antiparasitics [73]. Gigantol demonstrated binding to sterol 24-C methyltransferase with a binding energy of −6.9 Kcal/mol, and after applying gigantol to the parasite, it was determined that gigantol is able to inhibit it through the production of reactive oxygen species in a dose-dependent manner [73]. Further research should be conducted in animal models to determine the effectiveness of gigantol and other stilbenes as a treatment for leishmaniasis.

## 10. Combretastatin B-2

The non-prenylated stilbene combretastatin B-2 (Figure 11) was first identified by Pettit and Singh [74], who determined the compound in a sample obtained from the South African tree *Combretum caffrum*. They further elucidated the structure using NMR and then confirmed it through the synthesis of the compound. Combretastatin B-2 is not well characterized in comparison to other related compounds, including combretastatin A-1, A-2, A-3, A-4, and B-1, which have more extensive research and proven health benefits. Although combretastatin B-2 was first identified in 1987, it was not until over thirty years later that it was identified in cannabis by Guo et al. [16]. Combretastatin B-2 has also been identified in *Combretum psidioides*, a different tree in the same genus *Combretum*, where it was found to be present in an extract [75]. Although the extract was found to demonstrate antimycobacterial properties, further research should be undertaken to determine the contributions of isolated combretastatin B-2 or other compounds and their contribution to the observed findings [75].

### Anti-Cancer Effects

Guo et al. [16] investigated the anti-cancer effects of combretastatin B-2 against the cancer cell lines MCF-7, A549, HepG2, and HT-29. Combretastatin B-2 was moderately effective at inhibiting the proliferation of all four cell lines but had a stronger effect against MCF-7 and HT-29 than A549 and HepG2 [16]. The antiproliferative capability of combretastatin B-2 was not as high as other prenylated stilbenes that were tested, suggesting that the prenylated moiety is beneficial for anti-cancer potency. The cytotoxicity of combretastatin B-2 towards the four cancer types was limited in comparison to the positive control or other stilbenes tested [16]. These results suggest that although there may be some merit to investigating combretastatin B-2 as an anti-cancer agent, especially against breast and colorectal cancer, other stilbenes show more promise. When Pettit and Singh [74] first identified combretastatin B-2, they tested it against murine P388 lymphocytic leukemia cells. They demonstrated a significant inhibitory effect against these cells [74]. More research is needed to determine whether these findings translate clinically. 

## 11. 3-O-Methylbatatasin

The compound 3-hydroxy-5,4′-dimethoxybibenzyl (3-O-methylbatatasin) (Figure 12) was first discovered in the UV-C irradiated leaves of *Cannabis sativa* [76]. Marti et al. [76] investigated the effect abiotic stressors such as UV-C radiation could have on different plants, including cannabis. They measured the metabolome profile of these plants following exposure to UV-C radiation to determine which secondary metabolites in cannabis were induced or inhibited in response to the radiation. This was the first time 3-O-methylbatatasin was ascertained in cannabis, and it was determined that 3-O-methylbatatasin was induced more than any other compound in cannabis in response to UV-C radiation [76]. The induction of 3-O-methylbatatasin could be a protective mechanism against oxidative stress in the plant as stilbenes have known antioxidant capability, but analysis of the overall antioxidant activity of the cannabis extract yielded no significant difference between the control and irradiated extracts [76]. Further research is needed to determine if pure 3-O-methylbatatasin is an antioxidant and whether its presence can be detected in cannabis that has not been subjected to UV-C radiation. An isomer of 3-O-methylbatatasin, 3-hydroxy-5,3′-dimethoxybibenzyl (3′-O-methylbatatasin III), is more extensively researched with evidence showing anti-inflammatory, anti-allergic, and herbicidal potential in the compound [77,78,79]. The overall therapeutic benefits of 3-O-methylbatatasin have not been investigated; however, the benefits seen in the isomer provide the rationale for investigating this stilbene. 

## 12. In Silico Modeling

As summarized in previous sections, stilbenes found in cannabis have a plethora of potential therapeutic benefits, but more research is needed to understand the extent to which these stilbenes could be beneficial, including their mechanisms of action in the body. In order to further our understanding of these stilbenes and their metabolism, we performed in silico modeling of the compounds and summarized the relevant results. Predictive modeling is beneficial for determining possible drug–drug interactions, substrates and inhibitors of metabolic enzymes, and pharmacokinetic and pharmacological properties, amongst other outcome measures. 

### 12.1. Methods

Modeling was carried out using the ADMET Predictor™ 9.5 (Simulations Plus, Lancaster, CA, USA). The chemical structures for the fourteen cannabis-derived stilbenes were researched and then drawn using ChemDraw. The compounds were saved from ChemDraw and uploaded as SDF files into ADMET Predictor™ 9.5. The default settings for the metabolism, transporter, and PhysChem modules were used to classify the stilbenes as substrates and inhibitors of various proteins, including human cytochrome P450 (CYP) isoforms, and to predict activity, pharmacokinetic, and permeability data for the stilbenes. Pharmacokinetic data generated were based on the predictive modeling of a 10 mg immediate-release tablet and human liver microsomal clearance. Data outputs from these modules were imported into Excel for collation and analysis. It is important to note that predictions made by ADMET Predictor™ 9.5 have an inherent variability due to the use of mathematical models and assumptions. This should be taken into account when interpreting the results of the simulations.

### 12.2. Results

The ability of the cannabis-derived stilbenes to interact with CYP family enzymes in the liver is crucial for understanding both the metabolic pathways the stilbenes follow and the potential interactions they may have with drugs. As summarized in Table 1, the predicted interactions of the stilbenes with CYP isoforms can be determined using in silico modeling. Stilbenes show various degrees of substrate interaction with CYP enzymes (Table 1). 3-O-methylbatatasin is a predicted substrate for seven isoforms, more than any other stilbene, whereas HM1 and α,α′-dihydro-3,4′,5-trihydroxy-4,5′- diisopentenylstilbene are potential substrates for just two of the CYP enzymes. Every stilbene tested is an apparent substrate for CYP1A2, and no stilbene tested acts as a substrate for CYP2C8. The inhibitory influence of the stilbenes on various CYP isoforms was also predicted (Table 1). All fourteen of the cannabis-derived stilbenes were predicted to inhibit CYP2C9, meaning that there are also potential drug interactions with medications such as amitriptyline, celecoxib, clopidogrel, fluoxetine, losartan, piroxicam, and valproate, which are substrates of CYP2C9 [80]. These data can be used to guide future in vitro or in vivo research regarding stilbene interactions with these CYP enzymes. The substrate interactions of stilbenes with UDP glucuronosyltransferase (UGT) isoforms are summarized in Table 2. All fourteen stilbenes showed predicted substrate interactions with UGT2B15, while none were predicted to interact with UGT1A4 (Table 2). The high number of interactions with the UGT isoforms suggests that the stilbenes have the potential to undergo glucuronidation in the body. 

The breast cancer resistance protein (BCRP) is a crucial transporter for the absorption, distribution, and elimination of drugs. In healthy tissue, BCRP is expressed in the intestinal epithelium, liver hepatocytes, and renal proximal tubular cells [81]. BCRP is present in cancerous tissue and is involved in the resistance of cancers to multiple chemotherapeutic agents [81]. Co-administration of BCRP inhibitors with anti-cancer drugs increases the bioavailability of these drugs, potentially increasing their effectiveness [82]. Table 3 summarizes the predicted interactions of the cannabis-derived stilbenes and BCRP. Our analysis has shown predicted inhibition of BCRP by over half of the tested stilbenes (Table 3). 

Despite major progress in treating human immunodeficiency virus (HIV) with antiretroviral therapies, the disease still represents a challenge to the medical and scientific community. Multiple integrase strand transfer inhibitors have been approved as a therapy for HIV due to their ability to prevent HIV-1 replication [83]. HIV-1 integrase is the protein responsible for inserting the reverse-transcribed viral genome into the host cell. The two steps of HIV-1 integrase action are 3′-processing, in which a 3′ dinucleotide is removed, and strand transfer, in which the viral and host DNA are linked [83]. Disruption of either of these two steps is beneficial in halting viral replication in HIV. Figure 13 represents the potency of the stilbenes at inhibiting either HIV-1 integrase strand transfer or HIV-1 integrase 3′-processing. Our data show the activity of all the stilbenes against HIV, with HM3 showing the strongest inhibition of both steps and 3-O-methylbatatasin showing the weakest inhibition. The stilbenes were also compared to the predicted values of the positive control dolutegravir, a commonly used antiretroviral (Figure 13). The stilbenes showed comparable inhibition of HIV-1 integrase to dolutegravir, with many of the stilbenes being more effective at inhibiting HIV-1 integrase 3′-processing. These data lay the groundwork for further investigation into the antiretroviral properties of these stilbenes. 

Determining the percentage of a compound that remains unbound from blood plasma proteins may be beneficial for understanding the efficacy of drug treatment. The unbound drug is considered active and free to be distributed into the target tissues of the body. Our in silico predictions, seen in Figure 14 for the percentage of stilbenes that remain unbound in the plasma, are beneficial for understanding the pharmacokinetic and pharmacodynamic actions of the compounds. Dihydroresveratrol was predicted to have the highest percentage unbound at 11.734%, while the rest of the tested stilbenes had between 3.904% and 8.772% unbound (Figure 14). These data should help guide the rationale for drug development studies through the determination of the proportion of the stilbenes free to act on the target receptors or enzymes. 

Understanding and predicting pharmacokinetic properties of these compounds is beneficial to the future pharmaceutical development of these products. Table 4 summarizes relevant pharmacokinetic data for the fourteen cannabis-derived stilbenes using predictive modeling of a 10 mg immediate-release tablet and human liver microsomal clearance. The predictive data reveal high percentage absorption (>90%) for ten of the stilbenes and suggest potential bioavailability (>50%) for eight of the stilbenes. Predictive modeling can also provide information regarding the plasma concentration, the area under the concentration-time curve, clearance, half-life, and volume of distribution, as summarized in Table 4. 

In silico modeling also allows for predictions regarding the permeability of the stilbenes to membranes such as the skin, the jejunum, or the blood–brain barrier, as seen in Table 5. These data suggest that compounds such as α,α′-dihydro-3,4′,5-trihydroxy-4,5′-diisopentenylstilbene, HM2, and HM3 with high skin permeability could be effectively delivered through transdermal routes of administration. This provides the rationale for the use of topical pharmaceutical formulations to provide the therapeutic benefit that these compounds possess. On the other hand, compounds such as gigantol and combretastatin B-2 have lower skin permeability compared to other stilbenes but instead have higher jejunal permeability, suggesting that an oral route of administration may be more rational for the development of these compounds. These compounds also have the potential to act on the central nervous system, with ten of the fourteen stilbenes predicted to have high levels of blood–brain barrier permeability (Table 5). Lipinski’s rule of 5 is a good marker for drugs that will be potentially orally active in humans. Only three of the stilbenes violate one of the rules, and for each, the violated rule has a LogP greater than five. While Lipinski’s rule of 5 does not guarantee effective pharmacological activity, the lack of rule violations by the vast majority of stilbenes suggests favorable pharmacokinetics for these compounds that will allow for easier development of effective pharmaceutics.

## 13. Conclusions

In conclusion, fourteen bibenzyl stilbenes have been identified in *Cannabis sativa* so far, though with the number of compounds identified in the plant ever increasing, more are likely to be characterized over time. The fourteen stilbenes have various degrees of investigational research into their therapeutic potential, with some members, such as dihydroresveratrol and gigantol, being very well characterized due to their presence in other plant sources, while others have just recently been discovered. These compounds show promise as therapeutic agents that could be developed into novel medications for a variety of conditions with further research and development. Our in silico modeling of stilbenes will help guide future research into the pharmacokinetic, pharmacodynamic, and therapeutic potential of the compounds. Further research must be conducted regarding poorly characterized stilbenes and yet uncharacterized compounds to be discovered in *Cannabis sativa* to fully understand the therapeutic potential of these polyphenols.

## Figures and Tables

**Figure 1 pharmaceutics-15-01941-f001:**
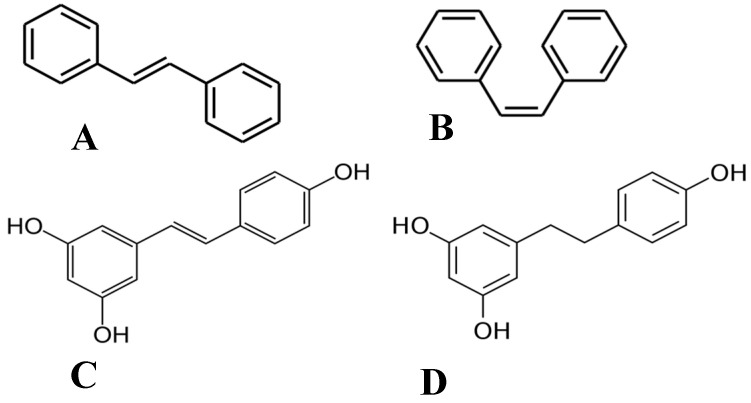
Chemical structures of (**A**) E-stilbene, (**B**) Z-stilbene, (**C**) stilbenoid (trans-resveratrol), and (**D**) dihydrostilbenoid (dihydroresveratrol).

**Figure 2 pharmaceutics-15-01941-f002:**
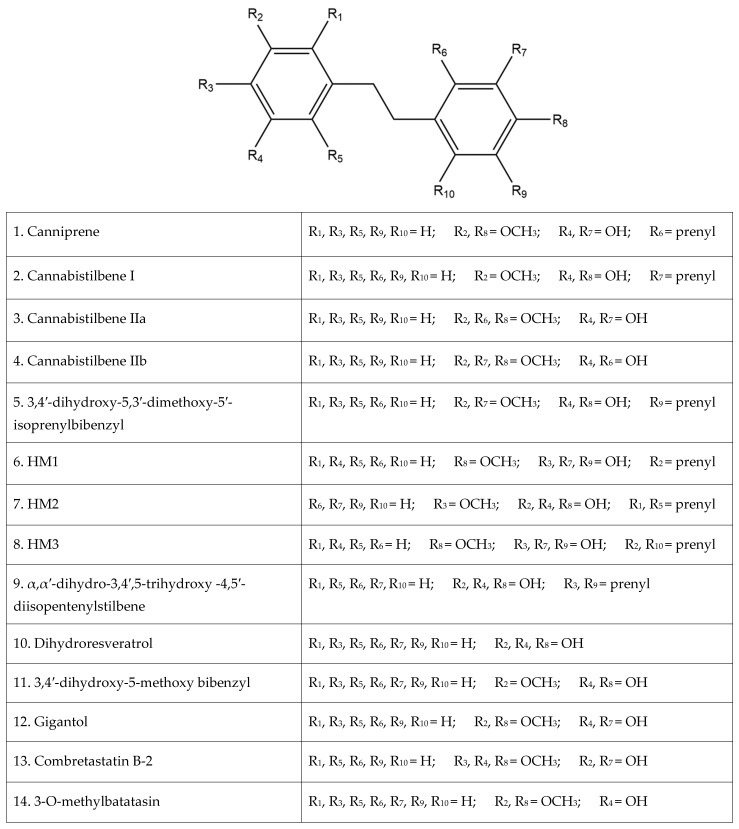
The fourteen identified stilbenes in cannabis, including a legend for the structural diagrams showing the functional group in each position along the bibenzyl structure.

**Figure 3 pharmaceutics-15-01941-f003:**
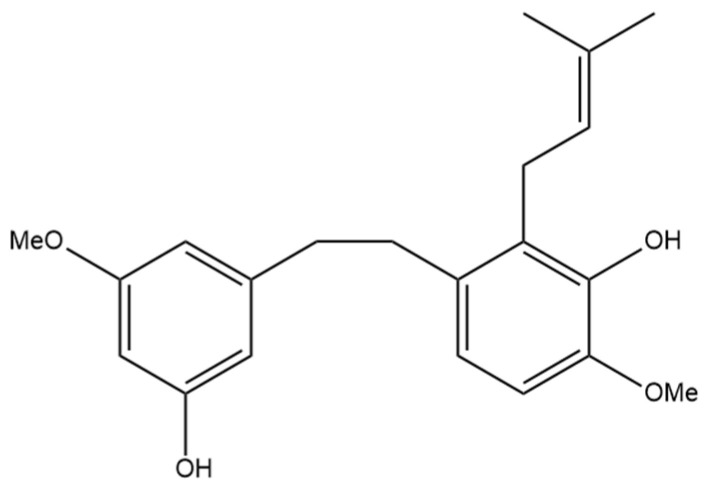
Structural diagram of canniprene (MW 342.43).

**Figure 4 pharmaceutics-15-01941-f004:**
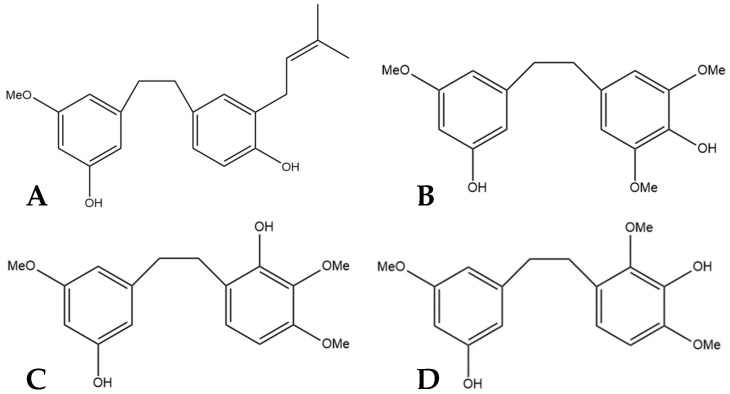
(**A**) Structural diagram of cannabistilbene I (MW 312.41). (**B**) An early possibility for the structure of cannabistilbene II. (**C**) Structural diagram of cannabistilbene IIa. (**D**) Structure of cannabistilbene IIb. (MW 304.34).

**Figure 5 pharmaceutics-15-01941-f005:**
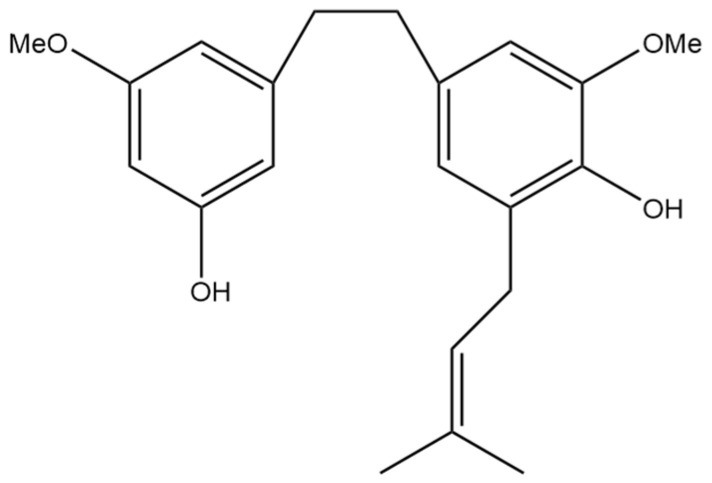
Structural diagram of 3,4′-dihydroxy-5,3′-dimethoxy-5′-isoprenylbibenzyl (MW 342.43).

**Figure 6 pharmaceutics-15-01941-f006:**
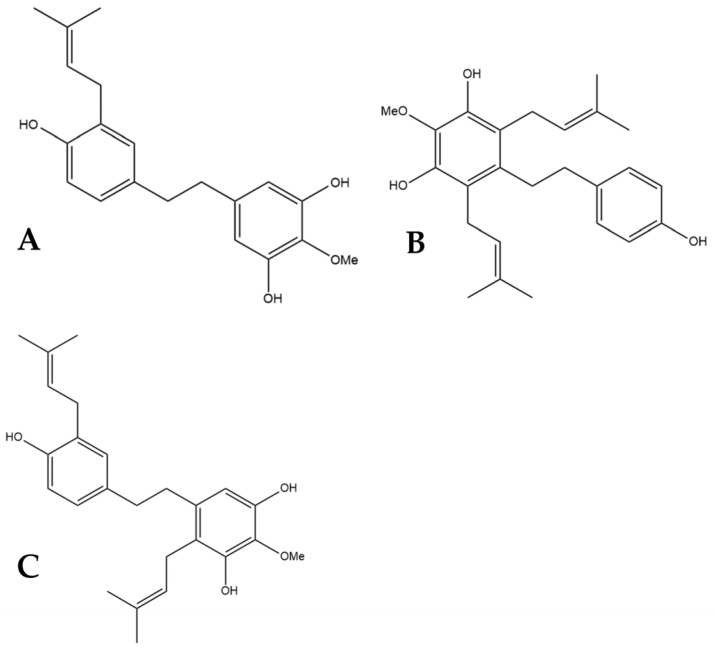
Structural diagrams of HM1 (**A**, MW 328.41), HM2 (**B**, MW 396.53), and HM3 (**C**, MW 396.53).

**Figure 7 pharmaceutics-15-01941-f007:**
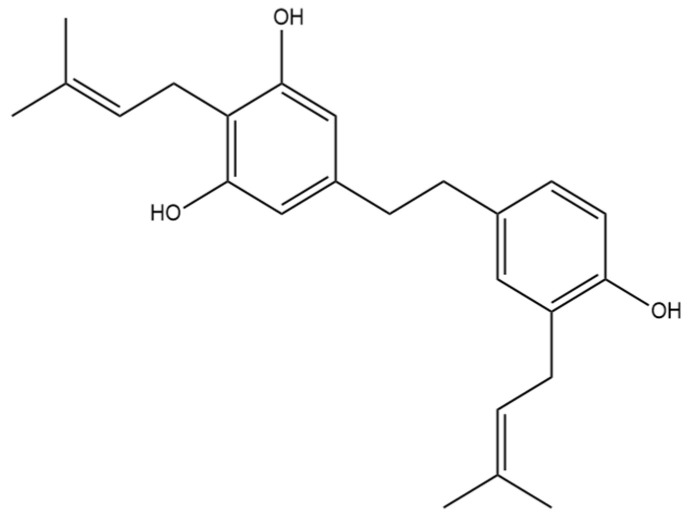
Structural diagram of α,α′-dihydro-3,4′,5-trihydroxy-4,5′-diisopentenylstilbene (MW 366.50).

**Figure 8 pharmaceutics-15-01941-f008:**
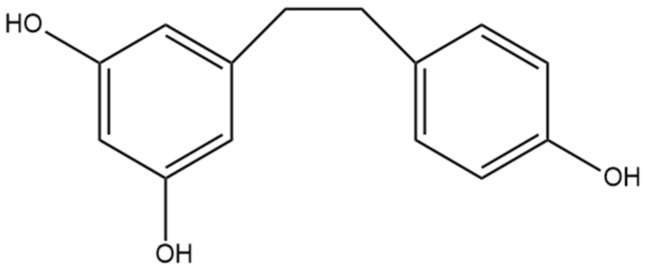
Structural diagram of dihydroresveratrol (MW 230.26).

**Figure 9 pharmaceutics-15-01941-f009:**
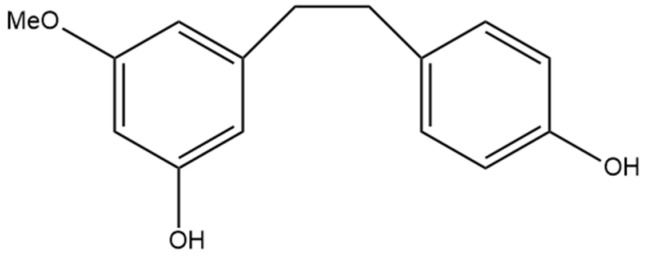
Structural diagram of 3,4′-dihydroxy-5-methoxy bibenzyl (MW 244.29).

**Figure 10 pharmaceutics-15-01941-f010:**
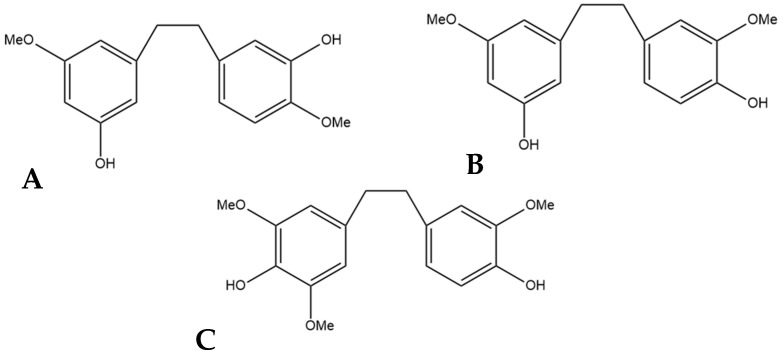
(**A**) 3,3′-dihydroxy-5,4′-dimethoxy bibenzyl (MW 274.32), the stilbene named gigantol found in cannabis. (**B**) 3,4′-dihydroxy-5,5′-dimethoxy bibenzyl, a common positional isomer of gigantol found in orchids. (**C**) 4,4′-dihydroxy-3,3′,5-trimethoxy bibenzyl, also occasionally referred to as gigantol.

**Figure 11 pharmaceutics-15-01941-f011:**
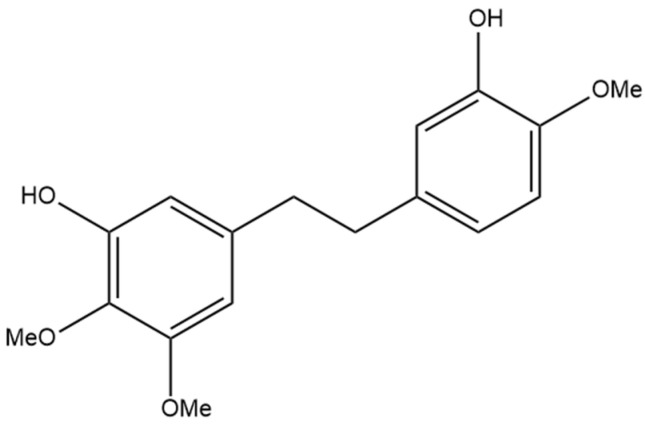
Structural diagram of combretastatin B-2 (MW 304.34).

**Figure 12 pharmaceutics-15-01941-f012:**
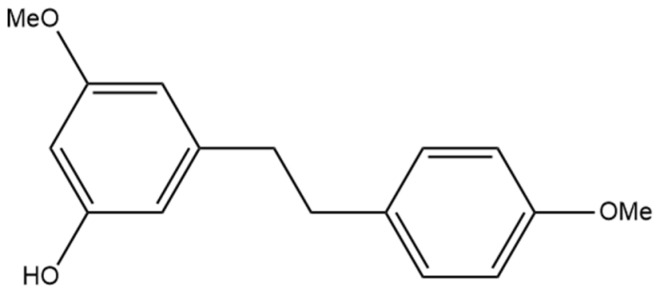
Structural diagram of 3-O-methylbatatasin (MW 258.32).

**Figure 13 pharmaceutics-15-01941-f013:**
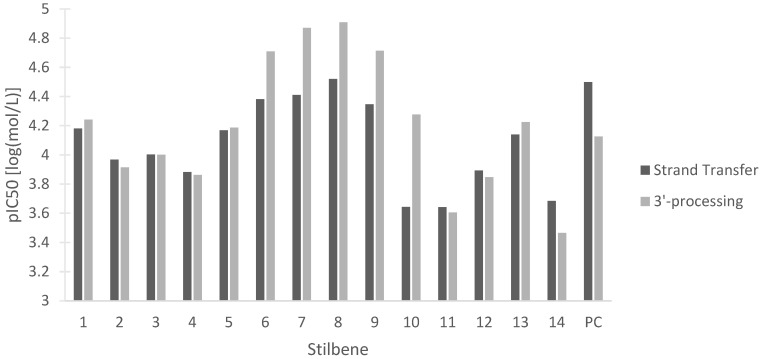
In silico pIC50 in log(mol/L) of 14 stilbenes in cannabis for the inhibition of HIV-1 integrase strand transfer and 3′-processing. (1) Canniprene; (2) cannabistilbene I; (3) cannabistilbene IIa; (4) cannabistilbene IIb; (5) 3,4′-dihydroxy-5,3′-dimethoxy-5′-isoprenylbibenzyl; (6) HM1; (7) HM2; (8) HM3; (9) α,α′-dihydro-3,4′,5-trihydroxy-4,5′-diisopentenylstilbene; (10) dihydroresveratrol; (11) 3,4′-dihydroxy-5-methoxy bibenzyl; (12) gigantol; (13) combretastatin B-2; (14) 3-O-methylbatatasin; (PC) positive control dolutegravir.

**Figure 14 pharmaceutics-15-01941-f014:**
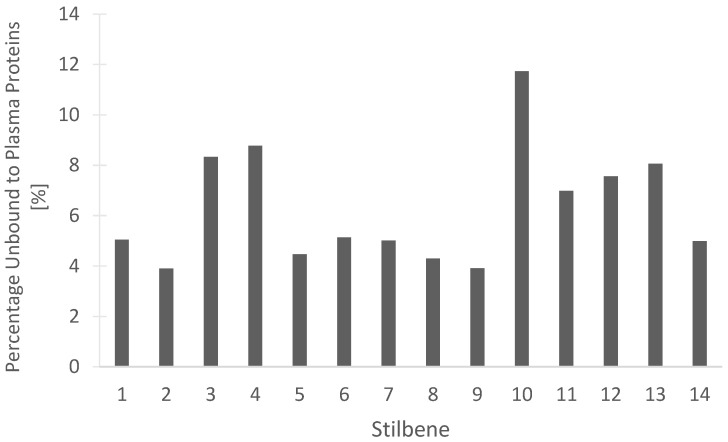
In silico prediction of the percentage of the 14 stilbenes in cannabis that are unbound to blood plasma proteins in humans. (1) Canniprene; (2) cannabistilbene I; (3) cannabistilbene IIa; (4) cannabistilbene IIb; (5) 3,4′-dihydroxy-5,3′-dimethoxy-5′-isoprenylbibenzyl; (6) HM1; (7) HM2; (8) HM3; (9) α,α′-dihydro-3,4′,5-trihydroxy-4,5′-diisopentenylstilbene; (10) dihydroresveratrol; (11) 3,4′-dihydroxy-5-methoxy bibenzyl; (12) gigantol; (13) combretastatin B-2; (14) 3-O-methylbatatasin.

**Table 1 pharmaceutics-15-01941-t001:** Heatmap for the predicted substrates and inhibitors for various CYP isoforms. Values indicate percentage confidence, where −99 is 99% confidence there is no interaction (shown in dark red) and 99 is 99% confidence there is an interaction (shown in dark green). (1) Canniprene; (2)cannabistilbene I; (3) cannabistilbene IIa; (4) cannabistilbene IIb; (5) 3,4′-dihydroxy-5,3′-dimethoxy-5′-isoprenylbibenzyl; (6) HM1; (7) HM2; (8) HM3; (9) α,α′-dihydro-3,4′,5-trihydroxy-4,5′-diisopentenylstilbene; (10) dihydroresveratrol; (11) 3,4′-dihydroxy-5-methoxy bibenzyl; (12) gigantol; (13) combretastatin B-2; (14) 3-O-methylbatatasin.

Stilbene	1	2	3	4	5	6	7	8	9	10	11	12	13	14
CYP SUBSTRATE														
CYP1A2	91	79	79	79	79	91	49	51	79	91	91	91	91	91
CYP2A6	−69	−66	−72	−67	63	−72	−86	−86	−98	−98	60	55	44	82
CYP2B6	−89	−67	−92	−89	−89	−92	−89	−89	−86	−92	−73	−92	−89	42
CYP2C8	−86	−83	−92	−92	−86	−83	−80	−86	−89	−80	−86	−92	−89	−89
CYP2C9	39	45	−67	−66	34	−67	37	33	−65	−85	45	−66	−66	50
CYP2C19	82	82	82	82	82	54	62	63	62	40	82	82	82	82
CYP2D6	−54	56	−60	−60	55	−63	−61	−61	−64	−95	−55	−56	−57	52
CYP2E1	−71	−69	63	63	−73	−71	−87	−82	−87	40	74	56	49	82
CYP3A4	73	−54	−42	−38	70	−62	72	−65	−48	−84	−84	−48	−51	−51
CYP INHIBITOR														
CYP1A2	−63	−79	−61	−57	−57	62	−61	−57	−97	−97	−97	−47	66	−76
CYP2C9	53	53	47	43	47	34	53	45	47	36	39	35	34	47
CYP2C19	−98	18	−99	−98	−99	−99	−99	−98	20	−84	−87	−98	−99	18
CYP2D6	51	51	−72	−76	51	44	70	70	55	−95	−60	−70	−84	−57
CYP3A4	59	−76	51	46	55	51	80	80	−65	−81	−90	−64	51	−78

**Table 2 pharmaceutics-15-01941-t002:** Heatmap for the predicted substrates for various UGT isoforms. Values indicate percentage confidence, where −99 is 99% confidence there is no interaction (shown in dark red) and 99 is 99% confidence there is an interaction (shown in dark green). (1) Canniprene; (2) cannabistilbene I; (3) cannabistilbene IIa; (4) cannabistilbene IIb; (5) 3,4′-dihydroxy-5,3′-dimethoxy-5′-isoprenylbibenzyl; (6) HM1; (7) HM2; (8) HM3; (9) α,α′-dihydro-3,4′,5-trihydroxy-4,5′-diisopentenylstilbene; (10) dihydroresveratrol; (11) 3,4′-dihydroxy-5-methoxy bibenzyl; (12) gigantol; (13) combretastatin B-2; (14) 3-O-methylbatatasin.

Stilbene	1	2	3	4	5	6	7	8	9	10	11	12	13	14
UGT SUBSTRATE														
UGT1A1	63	53	56	56	63	−45	56	−41	−66	−50	58	58	63	63
UGT1A3	90	90	51	51	90	97	97	97	97	83	64	64	73	−48
UGT1A4	−99	−92	−99	−99	−99	−99	−99	−99	−95	−99	−95	−99	−99	−86
UGT1A6	−97	−97	−97	−97	−91	−91	−97	−91	−83	48	−62	−83	−85	−80
UGT1A8	75	62	−66	−63	83	98	67	81	93	55	−46	72	83	−96
UGT1A9	97	97	97	97	97	97	76	97	97	97	97	97	97	97
UGT1A10	−65	−66	−77	−83	−83	53	−61	−56	53	50	−56	−61	−70	−90
UGT2B7	68	−60	68	72	72	93	85	93	−82	−55	−42	85	93	−44
UGT2B15	49	61	51	51	50	49	46	46	49	50	81	58	51	61

**Table 3 pharmaceutics-15-01941-t003:** Heatmap for the predicted breast cancer resistance protein interactions for the fourteen stilbenes in cannabis. Values indicate percentage confidence, where −99 is 99% confidence there is no interaction (shown in dark red) and 99 is 99% confidence there is an interaction (shown in dark green). (1) Canniprene; (2) cannabistilbene I; (3) cannabistilbene IIa; (4) cannabistilbene IIb; (5) 3,4′-dihydroxy-5,3′-dimethoxy-5′-isoprenylbibenzyl; (6) HM1; (7) HM2; (8) HM3; (9) α,α′-dihydro-3,4′,5-trihydroxy-4,5′-diisopentenylstilbene; (10) dihydroresveratrol; (11) 3,4′-dihydroxy-5-methoxy bibenzyl; (12) gigantol; (13) combretastatin B-2; (14) 3-O-methylbatatasin.

Stilbene	1	2	3	4	5	6	7
Substrate	62	64	79	79	62	79	−53
Inhibitor	64	67	−51	−51	62	58	64
**Stilbene**	**8**	**9**	**10**	**11**	**12**	**13**	**14**
Substrate	60	59	79	75	79	79	70
Inhibitor	67	81	−81	−49	−47	−44	52

**Table 4 pharmaceutics-15-01941-t004:** Pharmacokinetic data for the fourteen cannabis-derived stilbenes based on predictive modeling of a 10 mg tablet and human liver microsomal clearance. (1) Canniprene; (2) cannabistilbene I; (3) cannabistilbene IIa; (4) cannabistilbene IIb; (5) 3,4′-dihydroxy-5,3′-dimethoxy-5′-isoprenylbibenzyl; (6) HM1; (7) HM2; (8) HM3; (9) α,α′-dihydro-3,4’,5-trihydroxy-4,5′-diisopentenylstilbene; (10) dihydroresveratrol; (11) 3,4’-dihydroxy-5-methoxy bibenzyl; (12) gigantol; (13) combretastatin B-2; (14) 3-O-methylbatatasin. Fa is fractional absorption, Fb is the fraction bioavailable, Cmin is the minimum plasma concentration, Cmax is the maximum plasma concentration, Tmax is the time to maximum plasma concentration, AUC is the area under the concentration–time curve, AUCinf is the estimated AUC from time zero to time infinity, CL is the estimated total clearance, CLp is the plasma clearance, THalf is the estimated half-life, and Vd is the volume of distribution.

Stilbene	Fa (%)	Fb (%)	Cmin (ng/mL)	Cmax (ng/mL)	Tmax (h)	AUC (ng-h/mL)	AUCinf (ng-h/mL)	CL (L/h)	CLp (L/h)	THalf (h)	Vd (L)
1	88.6	49.13	3.42	5.28	13.5	92.27	166.45	29.52	29.52	14.9	634.34
2	99.55	55.34	3.24	6.79	5.51	109.95	185.3	29.86	29.16	16.55	713.22
3	99.99	71.23	2.1	40.61	2.24	339.19	354.11	20.11	20.1	4.95	143.68
4	99.99	69.1	1.05	44.69	2.12	314.15	320.06	21.59	21.58	3.94	122.58
5	85.07	44.91	3.1	4.49	14.71	79.34	143.54	31.29	31.29	14.27	644.03
6	96.57	48.67	2.6	6.06	8.97	96.61	141.31	34.44	34.45	11.82	587.55
7	66.53	25.88	1.62	1.69	19.72	29.8	97.06	26.66	39.83	22.32	858.52
8	51.18	12.25	0.68	0.69	20.49	12.33	53.76	22.78	49.5	27.32	898.04
9	98.08	45.23	2.31	4.18	7.91	70.32	124.22	36.41	34.25	17.33	910.28
10	100	78.71	2.97	51.28	1.73	450.04	472.66	16.65	16.65	5.22	125.3
11	99.99	83.29	10.68	29.92	2.05	451.62	674.32	12.35	12.33	14.25	253.93
12	99.99	76.17	4.88	35.63	2.56	386.09	437.16	17.42	17.39	7.26	182.56
13	99.99	67.54	1.97	31.14	2.62	275.16	289.86	23.3	23.27	5.24	176.22
14	99.98	78.29	8.13	16.05	3.17	271.65	513.56	15.25	15.14	20.47	450.24

**Table 5 pharmaceutics-15-01941-t005:** Predictive data related to the permeability of the fourteen cannabis-derived stilbenes. Lipinski’s rule of 5 score is defined as the number of times one of the following violations occurs: having a LogP above 5, more than 5 hydrogen bond donors, more than 10 hydrogen bond acceptors, or more than 500 Daltons molecular mass. (1) Canniprene; (2) cannabistilbene I; (3) cannabistilbene IIa; (4) cannabistilbene IIb; (5) 3,4′-dihydroxy-5,3′-dimethoxy-5′-isoprenylbibenzyl; (6) HM1; (7) HM2; (8) HM3; (9) α,α′-dihydro-3,4′,5-trihydroxy-4,5′-diisopentenylstilbene; (10) dihydroresveratrol; (11) 3,4′-dihydroxy-5-methoxy bibenzyl; (12) gigantol; (13) combretastatin B-2; (14) 3-O-methylbatatasin.

Stilbene	Permeability through Human Skin (cm/s × 10^7^)	Effective Human Jejunal Permeability (cm/s × 10^4^)	Blood–Brain Barrier Penetration (Confidence %)	Lipinski’s Rule of 5 Score
1	13.784	8.011	High (84%)	0
2	32.755	5.229	High (81%)	0
3	2.534	7.132	High (82%)	0
4	2.385	7.172	High (82%)	0
5	15.118	7.866	High (82%)	0
6	16.865	7.38	Low (39%)	0
7	31.767	10.02	High (70%)	1
8	47.737	8.562	Low (70%)	1
9	100.123	4.272	Low (45%)	1
10	4.824	3.951	Low (48%)	0
11	4.097	5.539	High (84%)	0
12	2.807	7.246	High (86%)	0
13	2.699	10.255	High (89%)	0
14	10.247	7.402	High (96%)	0

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
