# Peer review of "Therapeutic Potential and Predictive Pharmaceutical Modeling of Stilbenes in Cannabis sativa"

_pharmaceutics, 2023, doi:10.3390/pharmaceutics15071941_

Round 1

Reviewer 1 Report

Described in lucid English, the article is informative and useful. In page 2 line 48, the word 'dibenzyl' should be replaced by either 'dibenzylidenyl' or 'dibenzal'. In page 5 line 134, the word 'coronavirus' should be replaced by 'COVID'. 

English is fine, and lucid

Author Response

Reviewer 1
Thank you for your positive review, we have changed the nomenclature on page 2 line 48 to dibenzal and page 5 line 134 to COVID-19. 

Reviewer 2 Report

The manuscript called „Therapeutic Potential and Predictive Pharmaceutical Modeling 2 of Stilbenes in Cannabis sativa“ describes a very up-to-date topic.

The manuscript is designed well, however, for me, it was a bit confusing that the chapters were not organized always in the same order. Please, re-order the chapters in the same order, f.e. always start with describing anti-oxidant properties, anti-inflammatory etc.

For all tables, please, use the same background colors and the same word size.  

The abbreviations are not explained with the first using. Please, add the information throughout the whole manuscript.

The authors contribution has to be completed!

Data Availability Statement, Acknowledgments, Conflicts of Interest have to be added.

The language style is OK

Author Response

Thank you for your positive review and insightful comments, we have now structured the chapters so that they are not confusing and always organized in the same order with anti-inflammatory being first, followed by anti-cancer, followed by antioxidant should they all be present for a compound, and then following those are the compound specific sections. This in fact highlights where there is a lack of data or evidence of an effect for some compounds and where there is evidence for an effect.
The tables have been modified for greater consistency as suggested throughout, and all the abbreviations that were missing definitions have now been defined. Author statement and competing interest sections have also been added as suggested as well as data availability and acknowledgments.

Reviewer 3 Report

Timely review on therapeutic effects of secondary metabolites of cannabis that haven't been extensively studied. Minor editing will be beneficial to the manuscript.

One point that is still unclear after reading the entire manuscript is if the authors are suggesting that some of the therapeutic effect of cannabis could be explained by the effects of these metabolites or if the authors are suggesting extracting these metabolites from cannabis and study them to use them as separate medications. This should be clarified maybe in introduction and conclusion to open a close a necessary logical loop.

Figures:

- fig 1 has molecules A and B that are bigger than C and D. Moreover, the caption should state “Chemical structure of …”

- starting from figure 3, the authors are not citing any of the figures in the text as it needs to be.

- figure 4 should be merged with figure 5

Throughout the manuscript:

-        Especially in the first part of the manuscript, the authors are focusing on in silico and in vitro data, they should also add a sentence where they state if there are any preclinical evidence and if not, that are very much needed to prove the importance of these metabolites.

-        Line 296, the authors should translate the results they are reporting in a similar way of all the others reported in the text (μM instead of  x104 M-1)

-        Line 329 “CoA” shows up without specifying the not abbreviated version of it.

In silico modeling:

Not sure if the journal is allowing the authors to include original data in this review, but if the answer is yes then the authors should include a method section.

Moreover, the tables summarizing the in silico results is very hard to read, I suggest substituting them with heatmaps.

-fig 14 and 15. Why are there no error bars in these graphs?

References:

-        Line 866 ref 1 has an additional “1” before the first author and it appears to have a second set of names.

-        Some of the references don’t have the year in bold eg. #1; 14; 19. Please edit these.

Author Response

Thank you for the very helpful comments and positive review, a sentence in the introduction and conclusion has been added to create a logical loop with regards to the use of these compounds in medications outside of cannabis.
Figure 1 has been fixed so the images are the same size, and the caption makes more grammatical sense. The figures throughout the paper that were not being referenced are now referenced in the text and Figure 4 and Figure 5 have now been combined. The units for the numbers stated in line 296 have been changed to show consistency with other values and CoA has been defined on what was line 329.
A methods section has been added in section 12.1 of the manuscript. The tables that showed percent confidence measures have now been substituted with heatmaps for easier comprehension. We cannot provide error bars for the two bar graphs in the in silico section as the modeling provides us with exact predictions based on the database and no confidence measures, and there are no differences between repeat runs of the modeling, however we added to section 12.1 a description of the inherent variability that is present when interpreting the results of mathematical modeling.
The references that did not have the years bolded are now bolded, for reference 1 the source is a book, so the second set of names is there to credit the editors of the book.
Thank you for all the constructive feedback for this manuscript and we look forward to any additional comments you may have.